# Natural Language Processing to Extract Information from Portuguese-Language Medical Records

Naila Camila da Rocha [1,*] , Abner Macola Pacheco Barbosa [2] , Yaron Oliveira Schnr [2] , Juliana Machado-Rugolo [3] , Luis Gustavo Modelli de Andrade [2] , José Eduardo Corrente [4] and Liciana Vaz de Arruda Silveira [1]

1. Department of Biostatistics, Institute of Biosciences, Universidade Estadual Paulista (UNESP), Botucatu 18618-970, Brazil
2. Medical School, Universidade Estadual Paulista (UNESP), Botucatu 18618-970, Brazil
3. Health Technology Assessment Center (Clinical Hospital of the Botucatu Medical School), Botucatu 18618-970, Brazil
4. Research Support Office, Fundação para o Desenvolvimento Médico e Hospitalar (FAMESP), Botucatu 18618-687, Brazil
* Correspondence: naila.rocha@unesp.br; Tel.: +55-11940300705

**Abstract:** Studies that use medical records are often impeded due to the information presented in narrative fields. However, recent studies have used artificial intelligence to extract and process secondary health data from electronic medical records. The aim of this study was to develop a neural network that uses data from unstructured medical records to capture information regarding symptoms, diagnoses, medications, conditions, exams, and treatment. Data from 30,000 medical records of patients hospitalized in the Clinical Hospital of the Botucatu Medical School (HCFMB), São Paulo, Brazil, were obtained, creating a corpus with 1200 clinical texts. A natural language algorithm for text extraction and convolutional neural networks for pattern recognition were used to evaluate the model with goodness-of-fit indices. The results showed good accuracy, considering the complexity of the model, with an F-score of 63.9% and a precision of 72.7%. The patient condition class reached a precision of 90.3% and the medication class reached 87.5%. The proposed neural network will facilitate the detection of relationships between diseases and symptoms and prevalence and incidence, in addition to detecting the identification of clinical conditions, disease evolution, and the effects of prescribed medications.

**Keywords:** medical records; named entity recognition; neural networks

## 1. Introduction

Medical records are documents that contain sociodemographic information, health status, and the clinical history of hospitalized patients [1]. Data are collected from individuals upon entering a medical facility and are maintained until their discharge. These records include data on anamnesis, clinical examination, complementary exams, disease evolution, and prescribed medication.

Several studies have concluded that the analysis of these medical records can provide important information for decision-making in medical diagnoses [2–8]. However, they include a wide range of unstructured narrative data which involves some complexity in their analyses due to the use of acronyms, negation adverbs, and grammatical errors, and this data differs widely due to cultural differences and variations in descriptive and writing styles, along with the possibility of human error in filling out or typing the information.

Another challenge in the analysis of medical records is the existence of free text fields such as "patient evolution". This is a field in which many professionals choose to describe information that could be included in specific structured fields, making it difficult to search for specific data. Thus, the development of studies using unstructured narrative healthcare

information generally requires significant pre-processing by researchers, which can make these studies unfeasible.

Previous research has shown that the combined use of structured and free text data has led to the discovery of new relationships between diseases and symptoms, the identification of clinical conditions, and the identification of preventive actions for epidemiological outbreaks [9–11].

The primary advances in named entity recognition (NER) are in the English language, which has a wider range of available technologies in comparison to other languages such as Portuguese. The Portuguese language has limited resources, primarily with respect to specific contexts, such as healthcare.

Therefore, the aim of this study was to develop a neural network that uses unstructured data from medical records to obtain information regarding symptoms, diagnoses, medications, conditions, exams, and treatment. In this way, it will be possible to simplify the search for specific characteristics that do not appear in the structured data, improving the performance in daily searches for information at the Clinical Hospital of the Botucatu Medical School (HCFMB). This work presents an extraction of information from 1200 clinical texts in Portuguese, with manually annotated named entities, and it is one of the largest datasets in the literature.

This article is organized into five sections: (1) Introduction–presenting the problem, context, motivation, objectives, and contributions. (2) Results–describing how we present the socioeconomic and demographic data of the sample used, in addition to information regarding the type of care received, such as medical specialty and international classification of disease (ICD) description. Additionally, the outcomes of the model evaluation measures are provided and contrasted with the top outcomes from the Portuguese-language NER literature. The results of the cluster analysis are then presented, with an emphasis on the entities related to medications. (3) Discussion–providing examples of the model's classifications to show how it may learn and understand context. In addition, the interpretation of each group generated by the cluster analysis is summarized. (4) Materials and methods–showing the current literature plus samples of medical records used in the training, testing, and implementation of the model. The procedures used to build the model in the subsection titled "Methods" are outlined, and the corpus construction, the data categories to be retrieved, the text labeling tool, the convolutional neural network (CNN) architecture, certain features of the spaCy architecture, and the optimizers and hyperparameters presented in the model are described. The model's results post-processing to conduct a cluster analysis, fusing them with data from structured fields and demonstrating how the joint usage of this information might benefit the health area, are set out. (5) Conclusion–presenting the key insights, constraints, and suggestions for further research.

## 2. Results

### 2.1. Model Results

Of the 30,000 medical records in the sample, 13,823 (46.1%) were for male patients and 16,177 (53.9%) were for female patients. Patient age ranged from 0 to 104 years, with a mean of 45.6 years, median of 49.2 years, and standard deviation of 25.3 years.

For the female cohort, there was a more uniform distribution throughout the life cycle regarding the frequency of when they sought medical care. Males tended to seek medical care at ages close to birth and after 50 years of age as shown in Figure 1.

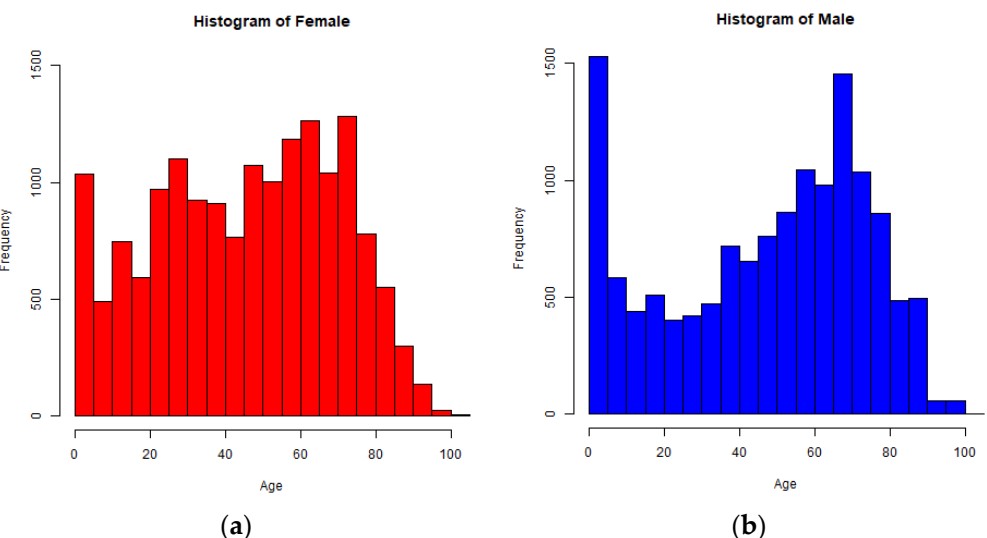

**Figure 1.** Age histogram by sex of the 30,000 patients in the sample. (**a**) Female age histogram of the sample. (**b**) Male age histogram of the sample.

Table 1 presents the demographic and socioeconomic variables that describe the sample.

**Table 1.** Demographic and socioeconomic variables of the 30,000 medical records.

| Factors | Categories | *n* | % |
|---|---|---|---|
| **Demographics** | | | |
| Sex | Male | 13,823 | 46.1 |
| | Female | 16,177 | 53.9 |
| Age group | 0–12 years | 4494 | 15.0 |
| | 13–19 years | 1431 | 4.8 |
| | 20–59 years | 13,730 | 45.8 |
| | 60 or older | 10,345 | 34.5 |
| Race | White | 26,348 | 87.8 |
| | Black | 1016 | 3.4 |
| | Mixed race | 2311 | 7.7 |
| | Asian | 39 | 0.1 |
| | Indigenous | 2 | 0.0 |
| | Unknown/did not declare | 284 | 0.9 |
| **Socioeconomics** | | | |
| Education | Illiterate | 3029 | 10.1 |
| | Basic literacy | 1993 | 6.6 |
| | Early primary | 4598 | 15.3 |
| | Primary | 10,225 | 34.1 |
| | Secondary | 6946 | 23.2 |
| | Higher education/undergraduate/postgraduate | 2586 | 8.6 |
| | Unknown/did not declare | 623 | 2.1 |
| Marital status | Single | 11,360 | 37.9 |
| | Married/common law | 13,031 | 43.4 |
| | Divorced | 2356 | 7.9 |
| | Widower | 3017 | 10.1 |
| | Unknown/did not declare | 236 | 0.8 |

This study was composed of patients treated by multiple clinical specialties. Table 2 presents the most common medical specialties seen by the patients in the sample.

**Table 2.** Descriptions of the most frequently seen medical specialties in the 30,000 records of the study.

| Medical Specialty | *n* | Value (%) |
|---|---|---|
| Internal medicine | 4099 | 13.4 |
| Pediatrics | 3886 | 13.0 |
| Internist | 3684 | 12.3 |
| General surgery | 1977 | 6.6 |
| Orthopedics/traumatology | 1319 | 4.4 |
| Nephrology | 1295 | 4.3 |
| Obstetrics | 1263 | 4.2 |
| Cardiology | 1165 | 3.9 |
| Clinical neurology | 1159 | 3.9 |
| Infectiology | 697 | 2.3 |
| Ophthalmology | 632 | 2.1 |
| Otolaryngology | 617 | 2.1 |
| Others | 8297 | 27.7 |

There was a wide variation in the International Classification of Diseases (ICD), the most prevalent of which occurred in only 1.8% of the medical records (Table 3).

**Table 3.** Descriptions of the most frequently seen ICD in the 30,000 study records.

| ICD Description | *n* | Value (%) |
|---|---|---|
| General examination | 527 | 1.8 |
| Acute pain | 513 | 1.7 |
| Other specified septicemias | 509 | 1.7 |
| Congestive heart failure | 486 | 1.6 |
| Unspecified acute myocardial infarction | 409 | 1.4 |
| End-stage kidney disease | 368 | 1.2 |
| Stroke not specified as hemorrhagic or ischemic | 337 | 1.1 |
| Status epilepticus unspecified | 325 | 1.1 |
| Other cerebral infarctions | 296 | 1.0 |
| Unspecified bacterial pneumonia | 294 | 1.0 |
| Eye and vision exam | 290 | 1.0 |
| Bronchopneumonia unspecified | 283 | 0.9 |
| Others | 25,363 | 84.5 |

*2.2. Data Quality Analysis and Model Validation*

The following results were obtained when we evaluated the exact matches, considering the identification of entity boundaries which were defined from the first to the last token (Table 4).

**Table 4.** Evaluation metrics for extracting named entities from the model.

| Entities | F-Score | Precision | Recall |
|---|---|---|---|
| Condition | 82.652 | 90.298 | 76.201 |
| Diagnosis | 49.272 | 54.424 | 45.011 |
| Exam | 54.664 | 72.609 | 43.832 |
| Medication | 80.966 | 87.474 | 75.360 |
| Symptom | 58.863 | 68.768 | 54.451 |
| Treatment | 47.312 | 57.592 | 40.146 |
| Model | 63.867 | 72.725 | 56.932 |

The disparity between the classifications of medical records in terms of the number of entities is discussed in Section 3.2.

### 2.3. Multivariate Statistical Methods—Cluster Analysis

Of the 28,800 medical records evaluated (the 1200 records used in the construction (training/testing) of the model were not considered), 17,382 contained information about drugs prescribed or used by the patient.

After calculating the silhouette width for clusters ranging from 2 to 10 for the PAM algorithm, we found that four clusters produced the highest values, indicating four as the ideal number of clusters in this case, as shown in Figure 2.

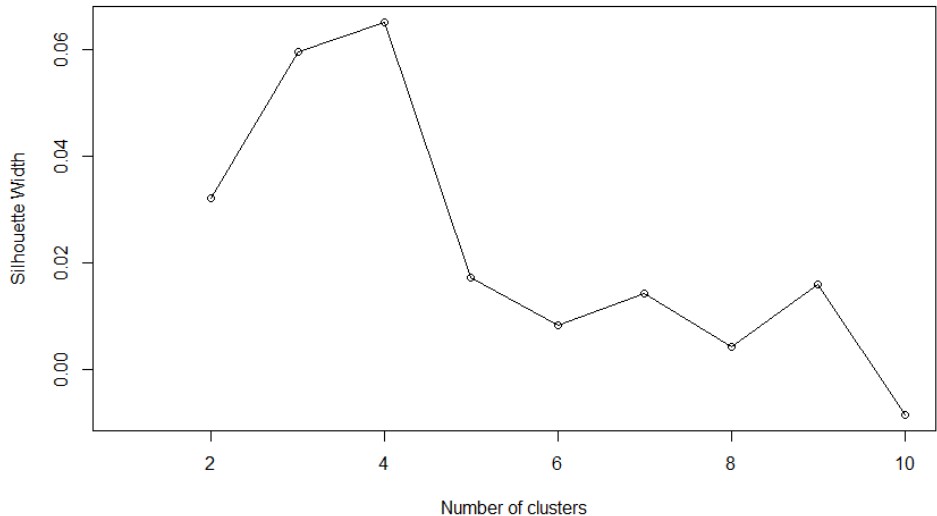

**Figure 2.** Silhouette width analysis showing the ideal number of clusters as four.

From the boxplot in Figure 3, we can see that Group 2 consisted predominantly of young people, whereas the other groups had a greater prevalence of elderly people.

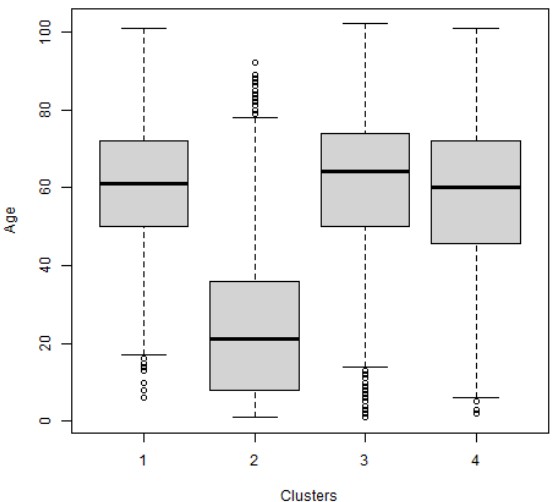

**Figure 3.** Boxplot showing age distribution for each cluster, considering all sample records except for those used for training and testing.

## 3. Discussion

### 3.1. Extracted Entities

Examples of the classifications made by the developed model are presented in Appendix A. All excerpts are taken from the "patient evolution" field of four records randomly selected among the 28,800 to which the model was applied.

In the first example, we have the successful extraction of the diagnosis and exam entities, even in cases where the names of the exams were abbreviated (i.e., "ct" for computed tomography and "xr" for X-ray).

In the second example, we can observe learning ability, with the model being able to detect different types of injuries as diagnosis, in addition to understanding the medications separated by the "+" sign and the abbreviation "physio" for treatment with physiotherapy.

In the third example, we can see the model's ability to detect condition and symptom alternately, showing efficacy in using the method in this type of analysis.

In the fourth example we can see the extraction of four out of the six entities from the model (symptom, condition, medication, and treatment). In the example, "ggc" is the abbreviation for good general condition.

### 3.2. Data Quality Analysis and Model Validation

With an F-score of 63.9%, the results were significant based on the relevant publications in Portuguese (detailed in Section 4.1), such as ClinPt [12], in which the exact match F-Score was 62.71%, and the "Named entity recognition for clinical Portuguese corpus with conditional random fields and semantic groups" [13] showed the best average F-Score, which reached 65% for the disorder class, with 60% for procedures and 42% for drugs, and which took into account the complexity of the database employed. However, it may be possible to improve the model's performance with a larger training database or adjustments to the hyperparameters. More information about the hyperparameters and optimizers used can be found in Appendix B.

When comparing the results of the initial training model of 200 records with the current one of 1200 records, there is significant improvement in precision. We also identified an imbalance in the number of entities in the exam and treatment classes present in the medical records, which interferes with our final results. This factor is explained by the average number of each entity per medical record: 2.8 reported conditions, 3.3 reported diagnoses, 1.0 prescribed or assessed exams, 2.4 reported medications consumed or prescribed, 4.4 reported symptoms, and 0.8 identified treatments.

### 3.3. Multivariate Statistical Methods—Cluster Analysis

After running the algorithm and selecting the number of clusters, a summary of the interpretation of each generated group is presented below.

Group 1—women with a mean age of 60.4, median age of 61.0, and standard deviation of 16.0 years. They had primary school education and were married. The predominant medications consumed were Aspirin, Losartan, Omeprazole, and Simvastatin.

Group 2—a mixed group (men and women) with a mean age of 24.6, median age of 21.0, and standard deviation of 20.5 years. They had high school education and were single. The predominant medications consumed were Dipyrone, Prednisolone, and Amoxicillin.

Group 3—men with a mean age of 60.4, median age of 64.0, and standard deviation of 18.7 years. They had high school/primary school education and were married. The predominant medications consumed were Aspirin, Simvastatin, Omeprazole, Furosemide, and Losartan.

Group 4—a mixed group (men and women) with a mean age of 58.0, median age of 60.0, and standard deviation of 19.5 years. They had high school/primary school education and were married. The predominant medications consumed were Norfloxacin, Clopidogrel, and Ciprofloxacin.

## 4. Materials and Methods

### 4.1. Related Works

In the literature [9–11], most studies using named entity recognition (NER) with semantic annotations are based on the English language, but some initiatives have been developed in other languages, such as Portuguese [12–27].

ClinPt [12] is a collection of 281 clinical case texts from the clinical journal Sinapse, published by the Portuguese Society of Neurology, with manually annotated named entities. The researchers used 20 clinical texts from the neurology service of the Centro Hospitalar Universitário de Coimbra (CHUC) in Coimbra, Portugal. Its relaxed match F-Score was 70.41% and its exact match F-Score was 62.71%.

A clinical corpus with the semantic annotations of various clinical narratives, including discharge summaries, outpatient records, nursing notes, and multiple clinical specialties, primarily in the areas of nephrology, cardiology, and endocrinology, was created using a dataset of 1000 texts from three hospitals [13]. The best average F-Scores reached 65% for the disorder class, 60% for procedures, and 42% for drugs.

One of the most well-known and recent semantically annotated corpora is Sem-ClinBr [14], which uses clinical texts from various medical specialties, document types, and institutions. The corpus has 1000 clinical notes and is labeled with 65,117 entities and 11,263 relationships, promoting research using secondary data from Portuguese electronic medical records.

The MedAlert Discharge Letters Representation Model (MDLRM), which contains 915 hospital discharge records in European Portuguese relating to hypertension, is another significant corpus referenced herein [15]. This system automatically collects data from free-text Portuguese discharge summaries. MedAlert achieved an entity recognition accuracy of 100% for entities such as anatomical sites, courses, and dates, and it achieved 93–99% for conditions, findings, and therapy. As a result of the recognition of active compounds, such as the recognition of insulin as a laboratory test, examining bodies are stated to have achieved an accuracy value of 69%.

The CORD-NER dataset uses comprehensive named entity recognition (NER) in the corpus COVID-19 Open Research Dataset Challenge (CORD-19) [16]. This CORD-NER dataset covers 75 refined entity types, common biomedical entities, and many new entity types explicitly related to COVID-19 studies. The researchers manually annotated over 1000 sentences for evaluation, and it achieved an F-Score of 77.28%.

Additionally, we can include recent studies [25–28], the majority of which suggested models based on bidirectional encoder representations from transformers (BERT) to enable the identification of named entities.

Finally, this work contributes to the creation of a corpus with 1200 clinical texts in Portuguese, with manually annotated named entities. It also presents the results of tests with different hyperparameters in order to optimize this type of model. A future version using BERT-based models to support the recognition of named entities is suggested in order to improve the results obtained.

*4.2. Data Presentation*

We used data from the Clinical Hospital of the Botucatu Medical School (HCFMB) in São Paulo, Brazil. The HCFMB is the region's largest public institution within the country's unified healthcare system (SUS) [29]. The hospital currently uses the SOUL MV electronic medical record system.

The study period was defined as 2012 to 2019, and in this period, the HCFMB registered 978,000 medical records. We randomly selected 30,000 of these records to compose our sample, representing 3% of the existing database. We excluded medical records for which the field "patient evolution" was left blank. The data was anonymized and no information regarding the identification of the patient or the medical team was collected.

We used structured information from the following fields: hospitalization date, sex, race, education, profession, blood type, outcome (death), birth date, marital status, medical specialty, and ICD. The unstructured data were extracted from the "patient evolution" field.

A total of 1200 entries were selected at random and added to the corpus. The model was later applied to an additional 28,800 records, for a total of 30,000 records in the sample. The results section presents the demographic and socioeconomic variables, the

ICD descriptions, and the most common medical specialties required by the patients in the 30,000 study records.

This study was approved by the Research Ethics Committee of the Clinical Hospital of the Botucatu Medical School in São Paulo, Brazil, on 20 May 2020 (protocol 4,038,578; CAAE 30365920.5.0000.5411).

Figure 4 shows more information about the overall project flow. First, we randomly selected 30,000 of these records to compose our sample. After that, 1200 of the 30,000 entries were randomly selected and added to the corpus. Data pre-processing was applied in order to ensure and enhance performance. This step is better described in Section 4.4. Then, data were manually classified as tuples and annotations were made in the spaCy annotation tool, as detailed in Section 4.5. These records generated a clinical corpus with the following entities: medication, condition, treatment, symptom, examination, and diagnosis. The next component in the flow is the holdout approach for the selection of the medical records used for training, which were reserved for testing. In this step, to optimize the model, we performed several simulations, changing the model's parameters. The hyperparameters that made the network reach the lowest loss value and that presented the best metrics for F-Score, recall, and precision were considered to be those with the best performance. Therefore, after training the model, the scorer.score function was used to verify the model's metrics against the test base. The optimized model was made available for internal use at the HCFMB and was also applied to the other 28,800 records (the 1200 records used in the construction of the model were not considered). We applied a cluster analysis to the generated base of medication, also considering the sociodemographic information present in the medical record, after the data post-processing stage (described in Section 4.7). Cluster analysis was applied to evaluate the extracted data and understand the potential to measure, explain, or predict the degree of relationship and impact of the variables, being an example of using the results of the model.

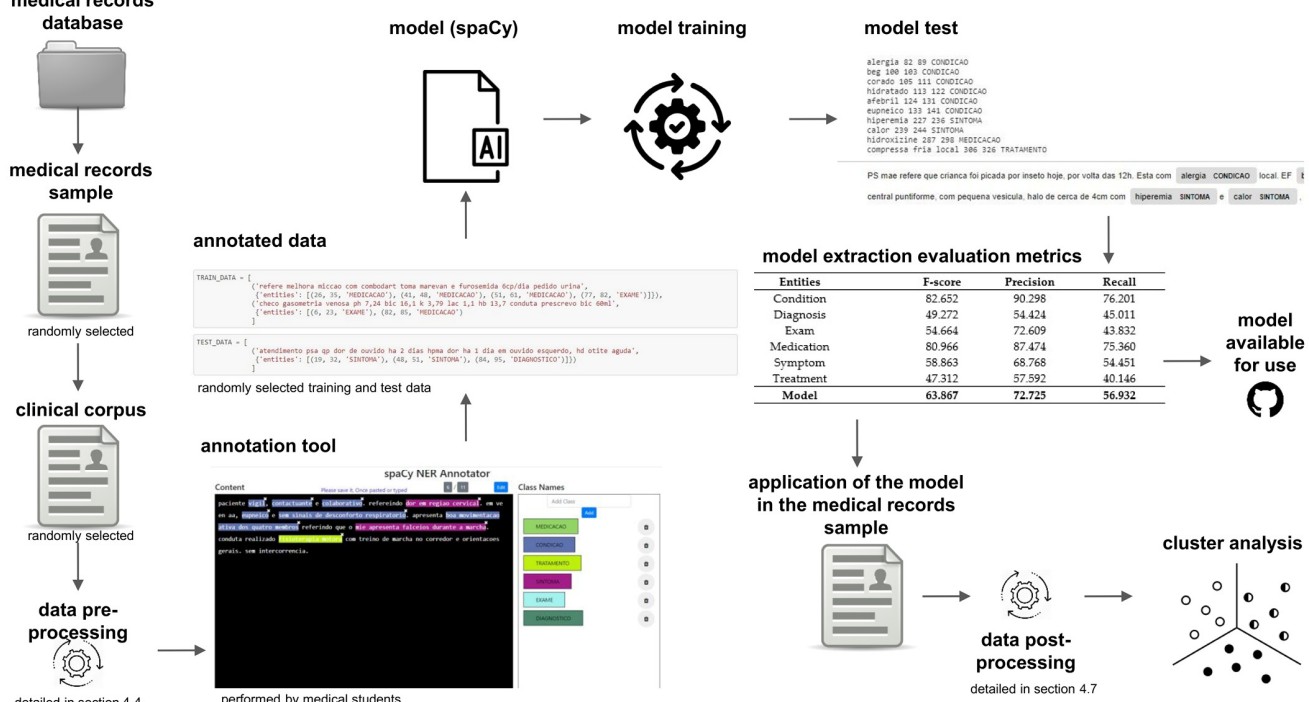

**Figure 4.** An overview of the project's flow.

### 4.3. Methods

To transform the data from the "patient evolution" field in the electronic medical record sample to a structured database, we built a model that used convolutional neural networks (CNN) for named entity recognition (NER).

We also applied natural language processing techniques in the model's pre-processing, database preparation, and post-processing to convert the result into a binary matrix (records/entities), indicating the presence of the entity in each record.

After data extraction, we used a cluster analysis with Gower distance and the partitioning around medoids (PAM) clustering method to demonstrate how the discovered data could provide relevant information for scientific/academic research and analysis.

### 4.4. Named Entity Recognition Tools

We used the spaCy library as it is an open-source library written in Python with native residual convolutional neural networks (CNN) (https://spacy.io/usage/processing-pipelines accessed on 3 October 2022).

Data pre-processing transformed the data into a format easily and effectively processed in data mining. In this stage, symbols, accents, line breaks, and special characters were removed and all text was standardized to lowercase letters. All tools are available on Github (https://github.com/nailarocha/ModeloNER accessed on 3 October 2022).

### 4.5. Clinical Corpus

Of the 30,000 medical records available for the study, 1200 were used in the construction of the corpus. When considering only single texts, without the repetition of standard texts and with the presence of entities, we found a total base of 1036 records. When applying the holdout approach to the 1036 medical records, 725 were used for training and 311 were reserved for testing. These records generated a clinical corpus with the following entities: medication, condition, treatment, symptom, exam, and diagnosis, which are defined as:

- medication: any medication in the medical record, whether prescribed in the current consultation or in the patient's history
- condition: previous conditions of the patient, including physiological characteristics considered normal (not symptomatic of any disease)
- treatment: previously prescribed or to be performed treatment related to diagnosis
- symptom: subjective phenomenon or physiological characteristic reported by a patient and usually related to a disease
- exam: medical procedure that aims to help diagnosis
- diagnosis: identification of disease from the descriptions of symptoms and the tests performed

The data were manually classified as tuples in the format "start_char, end_char, label", representing the position of the entities. These annotations were made in the spaCy annotation tool [30], which allowed for the assignment of custom labels in the text; thus, all existing entity intervals in these records were highlighted. At the end of the process, the gold standard data used to train/test the model were obtained. The classification was performed by medical students at the Júlio de Mesquita Filho Paulista State University.

The trained model was used to predict entities which were present in the test set and compare them with the gold standard. The results were returned to calculate the model's performance and evaluation using the precision, recall, and F1-score as metrics.

### 4.6. Model Description and Settings

The receiver of the tokenized sentence will be turned into a sentence matrix, the rows of which are word vector representations of each token. This is the first layer of processing that a convolutional neural network performs when processing text. Word embeddings handle every step of this procedure, and the filters combine the sentence matrix and produce the resource map. Each map is subjected to 1-max clustering, i.e., the highest number of each feature map is recorded. As a result, a feature vector for the penultimate layer is

created by concatenating the features from the univariate feature vector created from all the maps. The final softmax layer then takes this feature vector as an input and uses it to classify the sentence [31]. The CNN architecture for sentence classification is shown in Figure 5.

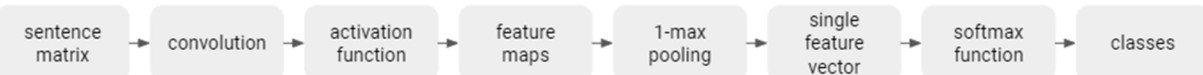

**Figure 5.** CNN architecture for sentence classification.

By using 1D convolutional filters on the input text, the spaCy architecture [32] predicted how nearby words could alter the existing entities, as nearby words can alter, reduce, or generate an entity. The input sequence was created by bloom embeddings, which modeled the characters, prefix, suffix, and part of speech of each word. The CNN used residual blocks, and a beam search was used to select the filter sizes.

To optimize the model, we performed several simulations by changing the model's parameters [33–36]. One of the simulations included the use of the minibatch function TRAIN_DATA, size = compounding (4.0, 32.0, 1.001), which generates an infinite series of composite values. With this function, we determined the start value, the stop value, and the compound, the latter being the composition factor for the series.

Different values were tested for the dropout hyperparameter (dropout rate). One of the alternatives evaluated was the use of the dropout decay function (0.6, 0.2, and $1 \times 10^{-4}$), which defines a high dropout rate at the beginning, then decreasing to a more reasonable value to avoid immediate network overfitting. Several authors have used a dropout of 0.5, and others recommend values close to 0.2 [33–36]. Therefore, the tests were performed to determine the values to be used in the model.

The optimizer defines how a model should change its weights to reduce the current error, which is calculated when the predicted data is compared to the actual results obtained. ADAM optimizers were tested with spaCy default settings [36] (learn_rate = 0.001, beta1 = 0.9, beta2 = 0.999, eps = $1 \times 10^{-8}$, L2 = $1 \times 10^{6}$, grad_clip = 1.0, use_averages = True, and L2_is_weight_decay = True) and the stochastic gradient descent (SGD; learn_rate = 0.001, L2 = $1 \times 10^{-6}$, and grad_clip = 1.0).

The other hyperparameters used were: beam_width = 1, beam_density = 0.0, beam_update_prob = 1.0, cnn_maxout_pieces = 3, nr_feature_tokens = 6, nr_class = 26, hidden_depth = 1, token_vector_width = 96, hidden_width = 64, maxout_pieces = 2, pretrained_vectors = null, bilstm_depth = 0, self_attn, depth = 0, conv_depth = 4, conv_window = 1, and embed_size = 2000.

*4.7. Data Post-Processing*

In this step, we created a table with the fields that occurred in the medical records, in addition to new columns created after applying the model. Each column was a class that was extracted (symptom, diagnosis, medication, exam, condition, and treatment) and filled with information extracted from the narrative field.

Subsequently, we performed a tokenization of the text, a process that involved the separation of each word into the text's constituent blocks, followed by conversion into numerical vectors. The objective was to obtain a matrix in which the extracted information was in columns.

Named entity recognition considers the context for the classification of entities, and it was possible to classify the text, even with acronyms, grammatical errors, and typographical errors. Therefore, we found 2578 unique words/phrases identified as medication. However, in order to proceed with the multivariate analyses, a manual review was necessary to remove repetition. As an illustration, Table 5 provides an example of the drug propranolol, which was input in 15 different ways. After reviewing and standardizing the database,

we obtained 1349 drug names that were considered as categorical variables in the cluster analysis detailed below.

**Table 5.** Example of the drug propranolol, which was written in 15 different ways in the sample of medical records.

| Number | Extracted | Correct Form |
|---|---|---|
| 1 | Popranolol | Propranolol |
| 2 | Porpranolol | Propranolol |
| 3 | Prapranolol | Propranolol |
| 4 | Pronalol | Propranolol |
| 5 | Propanalol | Propranolol |
| 6 | Propanol | Propranolol |
| 7 | Propanolol | Propranolol |
| 8 | Proparanolol | Propranolol |
| 9 | Propalol | Propranolol |
| 10 | Propanol | Propranolol |
| 11 | Propanolo | Propranolol |
| 12 | Propranolol | Propranolol |
| 13 | Proranolol | Propranolol |
| 14 | Prorpanolol | Propranolol |
| 15 | Prpranolol | Propranolol |

*4.8. Multivariate Statistical Methods—Cluster Analysis*

We applied a cluster analysis on the generated base of medication, considering the sociodemographic information present in the medical record, as discussed below:

- age (entry date − birth date)/365.25)
- sex (0 = female and 1 = male)
- race/color (0 = unknown/did not declare; 1 = white; 2 = mixed race; 3 = Black; 4 = Asian; and 5 = Indigenous)
- education (0 = unknown/did not declare; 1 = illiterate; 2 = basic literacy; 3 = first–fourth grade elementary school complete or incomplete (early primary); 4 = fifth–eighth grade elementary school complete or incomplete (primary); 5 = complete or incomplete high school (secondary); and 6 = complete or incomplete higher education, master's, or doctorate (higher education/postgraduate)
- marital status (0 = unknown/did not declare; 1 = single; 2 = married or common law; 3 = separated or divorced; and 4 = widowed)

The following variables were not directly considered in the cluster analysis: date of admission to the hospital, date of discharge, profession, blood type, outcome (death), date of birth, medical specialty that treated the patient, ICD, and international statistical classification of disease.

First, dissimilarities between the observations in the dataset were analyzed using the Gower distance due to the presence of mixed data. Then, the partition around medoids (PAM) algorithm was applied, a method that presented better results (and is the most suitable when the Gower distance is used).

The number of clusters was defined based on silhouette width analysis, a validation metric that determines how similar an observation is to its own cluster compared to the nearest neighboring cluster.

**5. Conclusions**

The results showed how the use of NER in clinical data can offer a powerful tool to support research studies, particularly for epidemiological work as this method manages to extract information that is not included in the structured fields of medical record databases.

The application of the model proved to be relevant and will support future studies that can be carried out to track the epidemiological profiles of the population served. Such

studies can provide more information to those responsible for managing resources so that resources are better applied, preventing waste and avoiding shortages of resources, medication, supplies, or professionals. In addition, by obtaining more information about medical care, it is possible to better prepare the healthcare system for epidemiological outbreaks or seasonal problems, thus mitigating their impacts.

Recent studies have proposed BERT-based models to support the recognition of named entities. These models will be taken into account as a potential upgrade to the model described in this work.

The number of annotators and reviewers is currently a constraint associated with the methods used. To improve the work, more annotators may be added, which would be another potential enhancement. Along with providing the model with feedback from other healthcare professionals, it is crucial to continually train the algorithm to be improved.

**Author Contributions:** Conceptualization, N.C.d.R.; formal analysis, N.C.d.R. and A.M.P.B.; data curation, N.C.d.R., A.M.P.B. and Y.O.S.; writing—review and editing, N.C.d.R., A.M.P.B., J.M.-R., L.G.M.d.A. and J.E.C.; supervision, L.V.d.A.S. and J.E.C. All authors have read and agreed to the published version of the manuscript.

**Funding:** This research received no external funding.

**Institutional Review Board Statement:** The study was approved by the Research Ethics Committee of the Hospital das Clínicas da Faculdade de Medicina de Botucatu/SP—Brazil (protocol code 4,038,578; CAAE 30365920.5.0000.5411 on 20 May 2020) and registered in SIPE (450/2019).

**Informed Consent Statement:** Not applicable.

**Data Availability Statement:** Not applicable.

**Acknowledgments:** This study was developed in the Health Technology Assessment Center of the Hospital das Clínicas—HCFMB, Botucatu, São Paulo, Brazil (LabData).

**Conflicts of Interest:** The authors declare no conflict of interest.

## Appendix A

Example 1

Portuguese version: Paciente encaminhada para avaliação de pseudoartrose DIAGNÓSTICO. tc EXAME e rx EXAME ok, com melhora de padrão prévio.

Translated to English: Patient was referred for pseudarthrosis assessment DIAGNOSIS. ct EXAM and xr EXAM ok, with improvement of the previous pattern.

Example 2

Portuguese version: Paciente com lesão osteocondral DIAGNÓSTICO, lesão de mm DIAGNÓSTICO e lesão deg DIAGNÓSTICO, onde tem dor SINTOMA. CD paracetamol MEDICAÇÃO + tramadol MEDICAÇÃO, fisio TRATAMENTO.

Translated to English: Patient with osteochondral lesion DIAGNOSIS, mm lesion DIAGNOSIS and deg lesion DIAGNOSIS, location of pain SYMPTOM. CD paracetamol MEDICATION + tramadol MEDICATION, physio TREATMENT.

Example 3

Portuguese version: Paciente consciente CONDIÇÃO, orientado CONDIÇÃO em tempo e espaço, descorada SINTOMA 1+/4+, hidratada CONDIÇÃO, acianótica CONDIÇÃO, anictérica CONDIÇÃO. Dispnéica SINTOMA, com cateter de O2 contínuo, mantendo saturação de O2 a 93.

Translated to English: Patient conscious CONDITION, oriented CONDITION in time and space, pale SYMPTOM 1+/4+, hydrated CONDITION, acyanotic CONDITION, anicteric CONDITION. Dyspneic SYMPTOM, with continuous O2 catheter, maintaining O2 saturation at 93.

Example 4

Portuguese version: PS mãe refere que criança foi picada por inseto hoje, por volta das 12 h. Está com alergia local SINTOMA. EF beg CONDIÇÃO, corado CONDIÇÃO, hidratado

CONDIÇÃO, afebril CONDIÇÃO, peso 16,1 kg, área central puntiforme com pequena vesícula, halo de cerca de 4 cm com hiperemia SINTOMA e calor SINTOMA em região do maleolo medial esquerdo. CD hidroxizine MEDICAÇÃO por 7 dias. compressa fria local TRATAMENTO.

Translated to English: ER mother said that the child was bitten by an insect today, around 12 pm. Child has local allergy SYMPTOM. PE ggc CONDITION, flushed CONDITION, hydrated CONDITION, feverless CONDITION, weight 16.1 kg, central puncture area with small vesicle, halo of about 4 cm with hyperemia SYMPTOM and heat SYMPTOM in the region of the left medial malleolus. CD hydroxyzine MEDICATION for 7 days. Local cold compress TREATMENT.

## Appendix B

The Figure A1 below shows an overview of the network performance by epoch in relation to the dropout function.

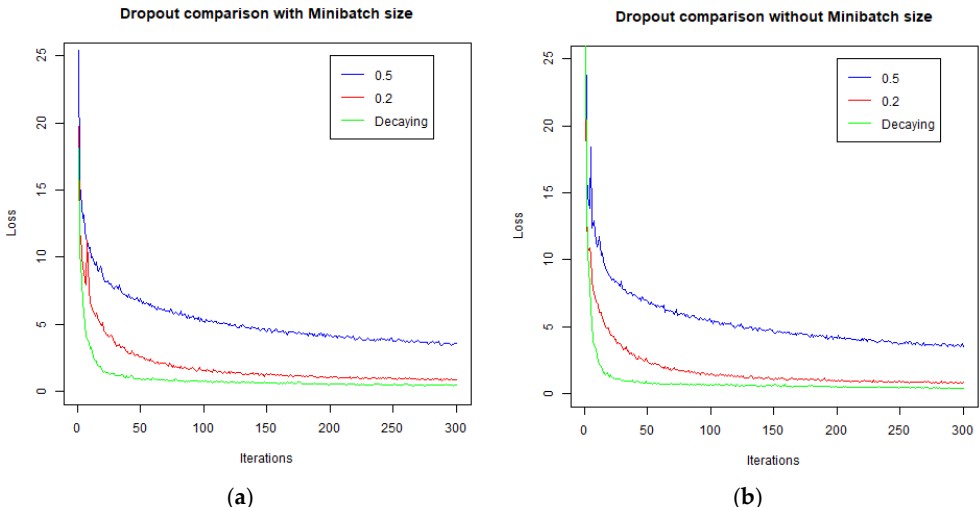

(**a**)　　　　　　　　　　　　(**b**)

**Figure A1.** Dropout and batch size hyperparameter simulation with 300 iterations. (**a**) Simulation considering decay with minibatch-size compounding functions for network regularization. (**b**) Simulation considering decay without minibatch-size compounding functions for network regularization.

When the dropout decay function $(0.6, 0.2, 1 \times 10^{-4})$ was considered, the lowest values of loss in both situations were obtained (with and without the minibatch-size compounding function), and from a certain point (approximately 35), the loss function showed an almost linear behavior. In contrast, the 0.5 configuration required a greater number of iterations to stabilize the model. It is also important to note that a dropout rate of 0.2 presented similar results to decay $(0.6, 0.2, \text{and } 1 \times 10^{-4})$, but it took slightly longer to stabilize the model. Another determining factor in the choice parameters was the training execution time for 1000 or more epochs, which was significantly higher than the others.

The ADAM optimizer with the hyperparameters recommended in the literature [35,36] presented better results than the stochastic gradient descent (SGD); therefore, it was used in the model.

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
