# Peer review of "Natural Language Processing to Extract Information from Portuguese-Language Medical Records"

_data_

Round 1
Reviewer 1 Report
The authors address an important issue related to the mining of information from medical data residing in EHRs both structured and unstructured data. There are a number of problems that need to be addressed.
1) The transformation of the EHR data using the CNN NER tool should be presented in detail so that the reader understands what is the new dimension of features introduced by this study.
2) The authors should explain why there are so much diversion in the recall and accuracy measures in the entities in Table 5. Why is there such a difference between condition, medication etc? Is this a problem of lack of information in the EHR, a problem with the transformation used, a problem of the model?
3) It is not clear how the authors chose the 1,200 EHRs to train their model. What was the profile of the patients> Was there an imbalance in structured data and unstrustured data in the 1,200 EHR set?
4) Following up the previous point, would a different 1,200 EHR set produce an analogous clauster analysis and different performance measures?
5) Please compare your study with analogous studies in the literature in using NLP and DL in mining information from EHRs and assessing diagnostics.
Author Response
Response to Reviewer 1 Comments
Point 1: The transformation of the EHR data using the CNN NER tool should be presented in detail so that the reader understands what is the new dimension of features introduced by this study.
Response 1: We have included more information in section 2.6. and in Appendix A to make this point clearer to readers.
Point 2: The authors should explain why there are so much diversion in the recall and accuracy measures in the entities in Table 5. Why is there such a difference between condition, medication etc? Is this a problem of lack of information in the EHR, a problem with the transformation used, a problem of the model?
Response 2: After your feedback, we pointed out below table 5, that this explanation can be found in section 4.2.: “This factor is explained by the average number of each entity per medical record: 2.8 reported conditions; 3.3 diagnoses; 1.0 prescribed or assessed exam; 2.4 medications consumed or prescribed; 4.4 reported symptoms; and 0.8 identified treatments.”
Point 3: It is not clear how the authors chose the 1,200 EHRs to train their model. What was the profile of the patients? Was there an imbalance in structured data and unstrustured data in the 1,200 EHR set?
Response 3: This point is presented in section 2.2.: “We randomly selected 30,000 records to compose our sample, representing 3% of the existing database. We excluded medical records for which the field “patient evolution” was left blank. We used structured information from the following fields: hospitalization date, sex, race, education, profession, blood type, outcome (death), birth date, marital status, medical specialty, and international classification of disease (ICD). The unstructured data were extracted from the “patient evolution” field.”
Point 4: Following up the previous point, would a different 1,200 EHR set produce an analogous clauster analysis and different performance measures?
Response 4: Cluster Analysis is a multivariate statistical procedure that attempts to group a data set into homogeneous subgroups. It is a mathematical technique that originates from revealing classification structures in real-world data. We did not use the 1,200 EHR in the production of the clusters, but the entire sample excluding the 1,200 that were used for training and testing. This point is detailed in section 3.2: "Of the 28,800 medical records evaluated (the 1,200 records used in the construction (training/testing) of the model were not considered), 17,382 contained information about drugs prescribed or used by the patient."
Point 5: Please compare your study with analogous studies in the literature in using NLP and DL in mining information from EHRs and assessing diagnostics.
Response 5: We have included the related works subsection to compare with analogous works, as suggested.

Reviewer 2 Report
The authors propose an algorithm based on ANN to extract medical record data.
The problem is well known but should be better formalized. What are the restrictions of the medical record data? What is expected from the algorithm for this problem? What is the acceptable accuracy level? How different is the problem for different languages? The formalization of the problem, requirements, constraints, etc. must be clearly described.
There are already some works on this topic. However, they are not referenced. A state-of-the-art should be provided with strong points and disadvantages of previous works, and how your solution improves it.
In section 2.2 you mention the application of NLP processing techniques for pre and post-processing. Which techniques? How?
A general flow should be described before starting the description of each individual step/tool so that the solution can be understood as a whole.
How do you should the vocabulary?
You mention the spaCy library. It should be explained and how it is used in the context of the entity recognition task.
What model did you use? The model was never described.
In the tests, what is the size of the training and evaluation datasets?
Why do you consider that the results are acceptable? Compared to what? Any constraints?
The work should be compared to other works from the literature. There are a few recent works with good results.
I suggest a reformulation of the paper. The authors should reorganize the paper, include a SotA, explain better the problem and the approach to solve it, the model, and a more extensive analysis of the results and comparison with the SotA.
Author Response
Response to Reviewer 2 Comments
Point 1: The authors propose an algorithm based on ANN to extract medical record data.
The problem is well known but should be better formalized. What are the restrictions of the medical record data? What is expected from the algorithm for this problem? What is the acceptable accuracy level? How different is the problem for different languages? The formalization of the problem, requirements, constraints, etc. must be clearly described.
Response 1: We revised the introduction and included more details of the work's motivation and contributions to the literature. We also include acceptable accuracy levels and the problem for different languages.
Point 2: There are already some works on this topic. However, they are not referenced. A state-of-the-art should be provided with strong points and disadvantages of previous works, and how your solution improves it.
Response 2: We have included the related works subsection to compare with analogous works, as suggested.
Point 3: In section 2.2 you mention the application of NLP processing techniques for pre and post-processing. Which techniques? How?
Response 3: Pre-processing techniques were better described in section 2.4 and post-processing in section 2.7.
Point 4: A general flow should be described before starting the description of each individual step/tool so that the solution can be understood as a whole.
Response 4: We have included this information in detail in Appendix A to make this point clearer to readers.
Point 5: How do you should the vocabulary ?
Response 5: I am sorry, I did not understand this question.
Point 6: You mention the spaCy library. It should be explained and how it is used in the context of the entity recognition task.
Response 6: It was better described in the second paragraph of section 2.6. Additional information can be found in the spaCy documentation, as the reference [36].
Point 7: What model did you use? The model was never described.
Response 7: We have detailed this information in the first paragraph of section 2.6 and Appendix A.
Point 8: In the tests, what is the size of the training and evaluation datasets?.
Response 8: This point is presented in section 2.5.: “Of the 30,000 medical records available for the study, 1,200 were used in the construction of the corpus. When considering only single texts, without repetition of standard texts and with the presence of entities, we found a total base of 1,036 records. When applying the holdout approach to the 1,036 medical records, 725 were used for training and 311 were reserved for testing.”
Point 9: Why do you consider that the results are acceptable? Compared to what? Any constraints? The work should be compared to other works from the literature. There are a few recent works with good results.
Response 9: We have included the related works subsection to compare with analogous works, as suggested.

Reviewer 3 Report
I would like to thank the authors for this work. The article presents an interesting application of NLP in the medical domain. However, please consider the points below in the next version.
(1)
The motivation behind the study needs further clarification. For example, is it based on a practical need for the university hospital? and/or the literature lacks such NLP studies in this context in particular?
Overall, the study should mention what it would add to the literature from a theoretical or practical aspect.
(2)
More recent related work is genuinely needed in the introduction. The related work should include studies that applied the state-of-the-art methods (e.g. BERT) to extract knowledge from free-text medical documents. For example:
https://doi.org/10.5220/0011012800003123
(3)
Please report the CNN architecture used in the study.
(4)
Please cite the references of dropout and ADAM.
(5)
Please use acronyms (e.g. NER, CNN) consistently throughout the manuscript.
(6)
Please discuss any limitations related to the methodology applied or other possible quality issues of the dataset.
(7)
The NLP research is currently dominated by the use of transformer models (e.g BERT). That said, I recommend considering that as part of the future work as well.
(8)
I find that the title is not positioning this work properly. It should be revised to position the article within the NLP context.
Author Response
Response to Reviewer 3 Comments
Point 1: The motivation behind the study needs further clarification. For example, is it based on a practical need for the university hospital? and/or the literature lacks such NLP studies in this context in particular?
Overall, the study should mention what it would add to the literature from a theoretical or practical aspect.
Response 1: We revised the introduction and included more details of the work's motivation and contributions to the literature.
Point 2: (2) More recent related work is genuinely needed in the introduction. The related work should include studies that applied the state-of-the-art methods (e.g. BERT) to extract knowledge from free-text medical documents. For example:
https://doi.org/10.5220/0011012800003123.
Response 2: We have included the related works subsection to compare with analogous works, as suggested.
Point 3: (3) Please report the CNN architecture used in the study.
Response 3: We have included more information in section 2.6. and in Appendix A to make this point clearer to readers.
Point 4: Please cite the references of dropout and ADAM..
Response 4: References [33 - 36] were better highlighted in the text.
Point 5: Please use acronyms (e.g. NER, CNN) consistently throughout the manuscript.
Response 5: We reviewed all acronyms as recommended.
Point 6: Please discuss any limitations related to the methodology applied or other possible quality issues of the dataset..
Response 6: Some limitations related to the methodology applied were included in the conclusion.
Point 7: The NLP research is currently dominated by the use of transformer models (e.g BERT). That said, I recommend considering that as part of the future work as well.
Response 7: The recommendation was included in the conclusion.
Point 8: I find that the title is not positioning this work properly. It should be revised to position the article within the NLP context
Response 8: Title updated to: "Natural Language Processing To Extract Data From Portuguese-Language Medical Records".

Round 2
Reviewer 1 Report
The authors addressed the revisions. The paper is improved.
Author Response
Thanks for all the suggestions. We have made additional improvements to the article.
Reviewer 2 Report
The paper has improved in general but still has some issues that need to be improved.
The state-of-the-art was extended. However, a final analysis is missing, that is, what can be improved and how it was improved with your approach.
The description of the model in section 2.6 is still unclear. Maybe a figure could help better understand the model. What is the size and complexity of the model?
You should include the project flow (Appendix A) in section 2.2 and explain it.
In section 4.2, you refer to an F-score of 63,9% and mention that it is significant given the relevant publications in Portuguese. Can you provide and refer to some of these publications, so that the reader can assess the value of your work?
Your work seems to be relevant, but you have to show that it brings added value to this research area.
Author Response
Response to Reviewer 2 Comments
Point 1: The state-of-the-art was extended. However, a final analysis is missing, that is, what can be improved and how it was improved with your approach.
Response 1: A paragraph with the final analysis has been included in section 2.1: “Finally, this work contributes to the creation of a corpus with 1200 clinical texts in Portuguese, with manually annotated named entities. It also presents the results of tests with different hyperparameters in order to optimize this type of model. A future version using BERT-based models to support the recognition of named entities is suggested in order to improve the results obtained.”
Point 2: The description of the model in section 2.6 is still unclear. Maybe a figure could help better understand the model. What is the size and complexity of the model?
Response 2: Figure added in section 2.6.
Point 3: You should include the project flow (Appendix A) in section 2.2 and explain it.
Response 3: The change was made as suggested.
Point 4: In section 4.2, you refer to an F-score of 63,9% and mention that it is significant given the relevant publications in Portuguese. Can you provide and refer to some of these publications, so that the reader can assess the value of your work?
Response 4: Publications were provided and referenced in section 4.2, as suggested: “With an F-score of 63.9%, the results were significant based on the relevant publications in Portuguese detailed in section 2.1, such as ClinPt [14] which the exact match F-Score was 62.71% and the "Named entity recognition for clinical Portuguese corpus with conditional random fields and semantic groups" [15] that the best average F-Score reached 65% for the disorder class, 60% for procedures and 42% for drugs, and taken into account the complexity of the database employed.”

Reviewer 3 Report
Thanks very much for accommodating the feedback. Just a minor comment to consider in the final version please. The new title is much better. However, I would just recommend replacing ‘data’ with ‘information’. This would position the work properly within the area of Information Extraction.
Author Response
Thanks for all the suggestions. We have made additional improvements to the article.
New title: "Natural Language Processing To Extract Information From Portuguese-Language Medical Records"
Round 3
Reviewer 2 Report
The authors have answered the issues identified by the reviewers and modified the article accordingly.